# Relationship between School Backpacks and Musculoskeletal Pain in Children 8 to 10 Years of Age: An Observational, Cross-Sectional and Analytical Study

**DOI:** 10.3390/ijerph17072487

**Published:** 2020-04-05

**Authors:** Tania López Hernández, Marina Caparó Ferré, Sílvia Giné Martí, Isabel Salvat Salvat

**Affiliations:** Department of Medicine and Surgery, Rovira i Virgili University, 43201 Reus, Spain

**Keywords:** musculoskeletal pain, child, schools

## Abstract

Back pain in children is a reality and various factors are involved in its etiology. The study’s aim was to analyze the relationship between the use and type of backpack and pain in children. An analytical observational cross-sectional study was conducted among 123 schoolchildren between 8–10 years. Data on the participants’ weight and height and their backpacks were collected, as well as the way of travel to school and their physical activity during the week. The results indicated that all backpacks were large because the backpack’s height is longer than torso length. Participants who studied in a traditional educational system (62.60%) carried backpacks that exceeded 10% of their body weight. Additionally, 31.7% of the students presented pain. There is no significant correlation between the weight or type of backpack and the pressure pain threshold collected from shoulders muscles. Participants who carried backpacks heavier than 10% of their body weight did not have more musculoskeletal pain or a lower pressure pain threshold than the others, although they did report greater fatigue. All these topics should be debated considering the student’s social environment and the backpack’s discomfort to the children, even though no relationship was found between musculoskeletal pain and backpack weight.

## 1. Introduction

Back pain in children is a reality because between 8% and 74% of children and adolescents have suffered from it at some point in their lives [1,2,3,4,5,6]; and it seems that various factors are involved in its etiology [1,2,3,7,8,9]. Excess weight in backpacks is considered one of these factors, along with age, gender, body mass index, peak pubertal growth, intense sports activity, and other psychosocial factors, as well as history of traumatic events or history of incorrect postural habits, sitting for extended periods, inadequate school furniture, and sedentary lifestyle [1,2,3,7,8,9].

The effects of carrying excess weight in school backpacks has been the subject of several publications, and there are great discrepancies in the reported results [10,11]. Transporting and carrying a backpack weighing more than the recommended maximum itself imposes a risk, and the risk to children and adolescents is increased since they are in the period of growth and maturation of the neuromusculoskeletal system [12]. Therefore, the weight of the backpack must not exceed 10%–15% of the body weight of the child or adolescent [3,4,7,8,12,13,14]. Exceeding this value causes posterior displacement of the center of gravity, consequently, excess tension is generated in the muscles of the back of the neck and the back, an effect that is associated with spinal column pathology [15] and decreased lung volume [16].

All these factors are related to musculoskeletal pain and hyperalgesia of the back muscles caused by the weight of backpacks [14].

Mwaka and others [14] consider that the areas most affected by carrying heavy backpacks are the shoulders and the cervical and lumbar spine; with cervical and shoulder pain related to the amount of weight carried, and pain in the lumbar area associated with the length of time for which the backpack is carried. For this reason, we know that the weight of the backpack falls directly on the shoulders and cervical area and they must exert a constant force to lift the shoulders and the scapula to counteract the force applied by the backpack [17,18]. The main muscles responsible for these actions are upper and middle trapezius, levator scapulae, and supraspinatus.

It is a known fact that algometry is a valid and reliable tool to assess hyperalgesia and musculoskeletal pain [19]. It is for this reason that muscles affected by carrying heavy backpacks such as upper and middle trapezius, levator scapulae, and supraspinatus are chosen for examination with algometry.

It has been shown that back pain is more frequent in children who walk to school or to an institute since they carry backpacks for longer periods of time [2]. To minimize the problems caused by carrying school backpacks, parents, teachers, and health professionals have attempted to find effective and functional alternatives. Currently, school lockers and/or tables with drawers are available in a large number of centers. This allows students to empty their backpacks and store the unnecessary contents to avoid carrying so much weight. Even so, it has been shown that much of the school population does not use these resources [3,7,12,20].

In addition to the short-term impact that carrying heavy backpacks has on children, there are data that associate pain in this population with the risk of suffering pain in adulthood [7,8,16].

The aim of the present study was to analyze the relationship between the use of various types of backpacks and pain, especially in the shoulder girdle muscles, in boys and girls between 8 and 10 years of age. We hypothesized that students who carried handled backpacks whose weight exceeded 10% of the student’s body weight would have more musculoskeletal pain and a lower pain threshold to pressure compared to students with backpacks less heavy or use a backpack with wheels.

## 2. Materials and Methods

### 2.1. Design of the Study

This was an analytical cross-sectional observational study whose protocol received the approval of the Clinical Research Ethics Committee of the Sant Joan de Reus University Hospital [14-09-18/9proj2]. Data collection was conducted in the primary schools Escola Joan Rebull (EJR) and InstitutEscola Pi del Burgar (IEPB) of Reus (Tarragona) during the months of October 2014 to March 2015.

### 2.2. Participants

The studied population consisted of schoolchildren between 8 and 10 years of age who attended two primary schools in Reus (Tarragona). These centers were selected because they use different teaching methods that affect the weight of students’ backpacks. While EJR uses textbooks and other materials provided by teachers, IEPB uses educational projects and the material used is less, only sheets and an agenda, making the students’ backpacks lighter.

The participants were eligible if (1) they were between 8 and 10 years of age (attending 3rd, 4th, or 5th grade), (2) they used backpacks to transport their school materials, and (3) their legal representatives had given written informed consent to their participation in the study. Students who (1) presented pathologies such as scoliosis or chronic fatigue or had any disability that made it difficult or impossible for them to transport their school materials, (2) had disabilities in communication or cognition, or (3) would not cooperate were excluded from the study.

### 2.3. Calculation of the Sample Size

Due to the lack of published literature about pressure pain threshold similar to this study, it was impossible to calculate the sample size; thus, 30 subjects were recruited from each of the two groups, as recommended in the literature [21]. However, estimating that the prevalence of pain was approximately 30% in this population [3] and assuming an accuracy of +/–8 percentage points at a confidence level of 95%, it was estimated that a minimum sample size of 126 children was required.

### 2.4. Variables

The key variables of the study were pain intensity, the pressure pain threshold, and the relationship between the participant’s weight and that of his or her backpack.

The intensity of pain was quantified using a Numerical Pain Scale (NPS). Scores on the NPS have been shown to be valid and reliable in many populations of children and adolescents as young as 8 years old, and perhaps even as young as 6 years old [22]. The subject was asked to indicate his or her degree of pain, related to pain intensity in any of the areas identified through the body map, using a horizontal 100-mm line that was divided into 10 equal parts; the left end of the line represented the absence of pain, corresponding to the value 0, and the right end of the line represented the worst pain imaginable, corresponding to a value of 10. The information was collected with the help of one of the researchers (T.L.). Pain intensity was classified as mild if it received a numerical score lower than 3.9 points, as moderate if it received a numerical score between 4 and 7.4 points, and as severe if it received a numerical score above 7.5 points [23,24,25].

The pressure pain threshold was assessed using an analogue algometer with a calibration certificate (Wagner Instruments FDX25 ForceGage, 25x0.02 Lbt). This method has strong reliability (Cronbach’s alpha ≥ 0.92) [26] and validity (convergent validity of pain thresholds across different body points was found to be high for all combinations assessed (Cronbach’s alpha values > 0.80)) [27]. The algometer consisted of a small cylindrical rubber (1 cm^2^) attached to a manometer. The units used were kg/cm^2^ (range of 0–10 kg with divisions of 0.1 kg). Three consecutive measurements were performed with the algometer held perpendicular to the skin. The average of the three measurements was considered the final value [28]. The pressure pain threshold was assessed bilaterally in four previously stipulated zones after asking the participants to indicate when the pressure began to cause pain. The analyzed areas were in the middle trapezius (between the first and the fourth dorsal vertebrae), the upper trapezius (angle of the neck), the supraspinatus (the middle portion of the muscular surface), and the levator scapulae (the angle of the neck immediately in front of the anterior edge of the trapezium).

The relationship between the weight of the participant and that of his or her backpack was assessed using the formula “weight of the backpack/weight of the participant x 100” to obtain the percentage. The weight of the patient and that of the backpack were measured using a scale (Orbegozo PB2210). This variable was used to divide the population into two groups according to whether the weight of the backpack was less than 10% of the student’s body weight or greater than or equal to 10% of the student’s body weight, as described in the literature [10,29,30,31].

The secondary variables were the location, duration and frequency of spontaneous pain, and lifestyle habits including the way to travel to school, physical activity during the week, etc.

The location of the pain was collected by means of a body map representing the front and back outline of a child; the participants were asked to point out the painful areas in their bodies using the map.

The duration and frequency of pain and lifestyle habits were assessed using a questionnaire based on various questionnaires found in the literature: Survey to Assess Backpack Use and Neck and Back Pain [2] and the questions used by others authors [3,4,7,9,12,13,16,20]. The first questions (1–11) refer to lifestyle habits, and the last (12–18) refer to pain (Appendix A).

### 2.5. Data Collection

Informative meetings were held with the parents of the participants to explain the characteristics of the study and its purpose and to resolve any doubts that might arise. During these meetings, the informed consent forms signed by the parents or guardians were collected.

Data collection for both groups was completed within a maximum period of 30 days after the meetings. In addition, to avoid changing the characteristics of the backpacks or the habits of the students, the intervention was conducted without prior notice of the day on which it would be held.

Once in the centers, the researchers explained to the participants the procedure that would be followed, and they then went to a large area provided by the schools. This was divided into 3 zones using opaque screens so that no evaluator was aware of the data collected by the others. A different evaluator was stationed in each area.

In zone 1, the first evaluator collected basic data on the participants to determine whether or not they met the eligibility requirements. If yes, they recorded the anthropometric data (student’s weight and height). The participant then went to zone 2.

In zone 2, the second evaluator recorded the intensity and location of the pain, interviewed the participant, and used his or her answers to complete the questionnaire on duration/frequency of pain and lifestyle (Appendix A). Upon completion, the participant went to the third zone.

In zone 3, the third evaluator performed the algometric measurements. To prepare for this, each participant was instructed using a standardized phrase: “You will feel a piece of rubber touch your skin. The pressure against your skin will increase little by little; at the moment that it starts to hurt, you will say “stop”. Next, with the subject in a sitting position lateral to the evaluator, the evaluator made three consecutive measurements, placing the algometer perpendicular to the participant’s skin, until the participant began to perceive pain [32]. This process was repeated three times in each of the eight evaluated zones. M.C. palpated the origin and insertion of the muscle in way to localize the middle of the belly’s muscle [26]: for the middle trapezius, the participant was asked to hug himself to slightly stretch the muscle, and the algometer was applied in a posteroanterior direction. To assess the upper trapezius, the algometer was placed in the angle of the neck with the child’s arms resting on the armrest of the chair and was applied against the index and middle fingers of the evaluator, who had clamped the muscle. In the same position, for the supraspinatus, the algometer was applied in the caudal direction, and for the levator scapulae, it was applied in the caudomedial direction. To facilitate the measurements, the participants were required to wear a tank top. The researcher in charge of performing the algometric evaluation did not know the other parameters of the participant.

At the end of the study, a reward was given to each of the students for their participation, and an informative talk was given about the correct use of backpacks.

### 2.6. Statistical Analysis of the Data

The quantitative values are presented as the mean and standard deviation. Given that some of the data do not follow a Gaussian distribution, other robust statistical measures such as the median and the interquartile interval are also used. The distributions of the categorical variables are described using absolute and relative frequencies.

Depending on whether or not the variable assumed a normal distribution, Student’s t-test or the Mann–Whitney U test was used for analysis of the association between the demographic values such as gender, student, and backpack weight or type of backpack, among others, of the two groups, which were defined according to the weight of the backpack in relation to the weight of the child (≥10% and <10%). For algometric values, only the Mann–Whitney U test was used since none of the variables assumes a normal distribution. Normality was tested using the Shapiro–Wilk test.

To find an association between the two groups and the variables of duration or frequency of pain and lifestyle habits listed in the questionnaire (Appendix A), the chi-square test or Fisher’s test was used, as appropriate. Finally, to analyze the correlation between the quantitative variables like weight or type of backpack and pressure pain threshold, the Pearson correlation coefficient with its corresponding *p*-value was used to analyze the significance of the correlation.

The statistical analysis was performed using SPSS statistical software. All comparisons were two-tailed, *p* values < 0.05 were considered to indicate statistical significance.

## 3. Results

### 3.1. Characteristics of the Population

A total of 145 participants were evaluated, of these, 22 were excluded when applying the eligibility criteria (16 for not having informed consent and six for being diagnosed with scoliosis). Therefore, the final analysis included 123 participants (47.97% girls) with a mean body weight of 32.66 (7.70) kg and a mean body mass index (BMI) of 16.62 with an interquartile range (IQR) of 15.44–19.12.

Of these, 77 (62.60%) carried backpacks that exceeded 10% of their body weight; all of these were students at EJR. The distribution of sex, as well as that of other variables, was equitable in these two groups (Table 1). The weight of the backpack and the proportion of the weight of the child it represents were different, as was the type of backpack used (Table 1). In general, backpacks with carrying handles were the most frequently used type (52.0%), especially in the <10% group; in the other group, the most frequently used type was backpacks with wheels. It was more common for the backpack to have two handles (only one participant had a backpack with one handle), and in 94% of the cases, the backpack was too large because the backpack’s height is longer than torso length [33].

Although 94.0% of participants live less than 15 min from school, only 56.9% of them walk (65.0% of the ≥10% group and 73.3% of the <10% group; Table 1).

Students whose backpacks exceed 10% of their body weight, unlike the rest, report frequently feeling tired when carrying their backpacks and believe that the backpacks weigh a lot. These students also leave the contents of their backpacks in the spaces they use most (Table 1).

A significant percentage (41.5%) of children reported that some member of their family had back pain, but no association was found with musculoskeletal pain in children. Although the observation displayed no statistical significance, there was a greater tendency for participants who carry backpacks weighing less than 10% of their body weight to have sedentary lifestyles; these students watch more television, play more video games, participate in sports less often and have, consequently, less cardiovascular demand (Table 1).

### 3.2. Pain

Of the participants in the study, 31.7% reported suffering musculoskeletal pain related to the use of the backpack. The mean pain intensity (in NPS) was 4.7 (1.8), and there was no difference between the groups. The majority of the subjects in the two groups presented moderate pain. A tendency of the <10% group to report longer duration of pain was observed; 33.0% of this group, in contrast to 16.7% of the ≥10% group, reported experiencing pain for longer than six months (Table 2).

With respect to the number of days per week that the children reported having pain, a statistically significant difference between the two groups was observed; most children in the ≥10% group reported rarely experiencing pain. A similar trend was observed in the duration of pain; the children in the ≥10% group reported shorter pain duration (Table 2).

Of all the participants, only one child in the <10% group had been absent from school because of this pain; 25.6% of the participants had consulted their family doctors about it (Table 2).

The areas of pain referred to by the participants, in order of prevalence, were shoulders (50.0%), interscapular (35.9%), cervical (20.5%), and lumbar (17.9%). There is a certain difference between the groups in that the children in the ≥10% group reported more pain in the shoulders (56.3%). On the other hand, the children in the <10% group reported more lumbar discomfort (29.2%). If the pain areas described by children who carry handled backpacks weighing more than 10% of their body weight are analyzed (this represents 16.3% of the total sample), the order of areas of pain remains the same, with shoulder pain reported by 30.0% and back pain reported by 0.0% (*p* = 0.05) (Table 3).

In an analysis by gender, it is observed that girls reported more intense pain (5.00 (1.81)) than did boys (4.54 (1.79)); the pain reported by girls was more frequent and longer-lasting, although none of these differences were statistically significant.

### 3.3. Pressure Pain Threshold

No statistically significant differences in pressure pain threshold were observed at any of the eight evaluated pain points (Table 4). On the other hand, the gender analysis shows that boys have higher values (between 0.51 and 1.65 kg/cm^2^) than do girls (between 0.27 and 1.22 kg/cm^2^), and the difference is statistically significant (*p* = 0.000).

There is no significant correlation between the weight or type of backpack and the pressure pain threshold of the points analyzed, although there is a moderate relationship between the presence of a low pain threshold and the existence of pain in the analyzed area; specifically, the algometry of the upper and middle trapezius is related to cervical pain (r = 0.447; *p* < 0.005), that of the supraspinatus is related to cervical pain (r = 0.482; *p* < 0.005) and to pain in the shoulder (r = 0.351; *p* < 0.005), and that of the fibers of the middle trapezius is related to low back pain (r = 0.426; *p* < 0.05).

It is also observed that all pressure pain thresholds of the evaluated points are moderately or strongly correlated. A low pain threshold at a muscle point is associated with pain in the contralateral muscle (0.701 ≤ r < 0.746; *p* < 0.005), and a low pain threshold of the upper trapezius fibers is related to low mean values for the middle trapezius of the same side (0.595 ≤ r < 0.697; *p* < 0.005). In addition, a correlation between the pressure pain threshold of a given muscle with the pain threshold of muscles on the same side (0.447 ≤ r < 0.685; *p* < 0.005) and with that of the contralateral muscles (0.215 ≤ r < 0.661; *p* < 0.005) was found.

## 4. Discussion

This is the first study that has aimed to relate the type and weight of student’s backpacks to their pressure pain thresholds. It is also the first study to include among its participants, schoolchildren who, when working on projects, do not carry backpacks that are too heavy. This has allowed a comparison of the two groups and an analysis of the effects of the transported weight on back and shoulder pain.

The sample used in this study is representative of Spanish schoolchildren in terms of BMI and carried weight. Indeed, the participants in this study have a mean BMI of 17.38 (2.74), similar to the value reported in similar studies; the mean value is slightly higher than that found by Dockrell et al. (2015) [11] and similar to that described by Arriba et al. (2016) [34], as expected for a sample like ours, which was obtained in a city of approximately 110,000 inhabitants. The age of the students in our study, approximately eight years, is somewhat lower than that of students in other studies [4,9]; this may perhaps explain why the average weight of the backpack relative to the weight of the child, for students whose packs exceeded 10% of their body weight was greater than that described in previous studies. However, the majority of the children in this study, 74.0%, used wheeled backpacks. This result contrasts with the results of Talbott et al. (2009), Yamato et al. (2018) [35,36], Alberola et al. (2010) [4], and Conti et al. (2010) [7], who reported that only 10% of students use wheeled backpacks. Our findings are favorable given that the literature recommends their use as an alternative to handled backpacks [10]. In our study, we collected data on the reasons for this change; the reasons seem more related to the fact that a wheeled model was considered more “trendy” and, in addition, offered a means of avoiding the pain caused by carrying a handled backpack.

This study also assessed the size of the backpacks and the carrying time. Thus, it was determined that 99.3% of children use backpacks that are too large relative to their bodies; if the backpack’s height is longer than torso length, it touches the head of the child and prevents its movement, and if the wheels are too high, the handle does not remain well adapted, causing increasing stress on the shoulder and back muscles [33]. Mwaka et al. (2014) [14,33] warn of the importance of using a properly sized backpack that is comfortable to reduce the risk of injury, in particular, these authors describe a higher incidence of lower back pain in children who transport backpacks that are too large and heavy for more than 20 min. This finding was not corroborated in our study because no child with a heavy backpack typically carried it for more than 20 min.

Regarding carrying time of the backpack, 94.0% of participants live a 15-min or shorter walk from their schools; therefore, the time during which they would carry their backpacks, if they walked, would be similar to that reported in other studies (Adeyemi et al., 2014 [29]; Dianat et al., 2013 [30]; Talbott et al., 2009 [35]). However, 41.5% of students arrive at their schools by car or public transport, a fact that could be directly related to the weight of their backpacks, given that the percentage of children in the <10% group who walk to school is slightly higher.

The prevalence of musculoskeletal pain (31.7%) found in this study falls between the values described in the literature, although it is lower than the 60.0% reported in similar studies [4,11]. This discrepancy may be due to the fact that in our study, only 16.0% of the children who had backpacks with handles carried backpacks that exceeded 10% of their body weight, while in the study by Dockrell et al. [11] this proportion was 77.2% (Alberola et al. [4], 2010). In agreement with those studies [4,11,36], little or no relationship was found between carrying the backpack (weight, size, or time) and the pain reported by the participants. In addition, the analysis of the hyperalgesia of body zones conducted in the present study in relation to carrying backpacks showed in principle that this relationship does not exist.

The pain described should be considered moderate. The average NPS score was 4.72 (1.8) points, and pain was reported to occur rarely (<1 day/week). These data are very similar to those reported by Alberola et al. (2010) [4]. Unlike that study, the location of pain was assessed in this study. The area in which pain was most often described was the shoulders (50.0%), followed by the interscapular area (35.9%). The location of the pain found in our study, which agrees with the location reported by Dockrell et al. (2015) [11], Dianat et al. (2014) [30], and Moore et al. (2007) [37], is a finding of interest since it indicates that pain associated with the use of backpacks is located more frequently in the shoulder girdle than in the lumbar area; much of the previous literature on the relationship between backpacks and pain focuses on the latter type of pain. In addition, we found that there was a higher prevalence of shoulder pain in the ≥10% group, while the <10% group reported more back pain. This finding is consistent with the literature, especially with an excellent systematic review published by Yamato et al. 2018 [36] which include 69 studies (*n* = 72,627 participants), that associates shoulder pain with the carrying of backpacks and suggests that lumbar pain is associated with psychosocial factors.

We found a very low rate of school absenteeism, only one student among those assessed had been absent from school due to musculoskeletal pain, which represents a percentage similar to the 97.0% that was previously described [4,36]. However, the percentage of participants who reported consulting their family doctors because of pain (26.0%) is much higher than that found in the literature (17.8%) [4].

The role of gender in musculoskeletal pain secondary to the use of backpacks is controversial. In the present study, girls reported more musculoskeletal pain than boys, although the difference was not significant. These results agree with those of Adayemi et al. [29], Dianat et al. [30], and Dockrell et al. [11], who studied populations of similar age to ours. On the other hand, authors [13,38] who have studied older populations do report a greater intensity of pain in girls, which they associate with the beginning of puberty that differentiates between genders.

The assessment of pain is subjective and is influenced by the character, environment, and previous experiences of the students [10]. The student’s perception of the weight of the backpack can also be influenced by these factors [10]. For these reasons, a question was included about whether the parents suffered from back pain because, although the point is controversial, a certain association between the two factors has been reported in the literature [10,11]. However, this relationship is non-existent in the sample analyzed in our study. Other aspects of a psychosocial nature were not investigated in the present study; this represents a limitation since the researchers noted certain differences in the participants from the two schools.

Indeed, the differences that were found between the two groups with respect to pain show a tendency of the group that carries less weight to report equal or greater pain than the group that carries heavier backpacks. Although they were not statistically significant, these differences could perhaps be related to the differences in the children’s natures; the students who carried more weight generally had a more active lifestyle, whereas students in the <10% group were more calm and organized and displayed a greater tendency to have a sedentary lifestyle (they watch television more, they play more video games, and they participate in sports less often than the participants in the other group). This fact may influence their perception of pain and of the weight of the backpack, although there is controversy in the literature on this point. Some authors have found no significant relationship between exercise and back pain in children [39], whereas Dockrell et al. (2015) [11] observed a similar association between passivity and pain in children: there is a higher prevalence of lower back pain in children who have more passive natures and who report being tired without a specific reason [11]. In summary, the children who carry more weight also play more sports, while those in the ≤10% group dedicate more hours to activities of a cultural and inactive nature. This could also make them more susceptible to nociception and have other harmful effects, apart from the effects of physical exercise on the modulation of nociception [40].

Concerning the algometric values, the eight muscle areas studied had lower pain threshold values (and therefore more pain) in the participants in the <10% group, although the differences were not statistically significant.

With the exception of the levator scapulae, in which pain was not found in previous studies, the values found in this study are very similar to those described in the literature in children for all the muscles analyzed [26,41,42,43,44]. In adults, it has been established that the cut-off point at which a zone can be considered hyperalgesic is 4 kg/cm^2^ [28], but there is no literature that establishes a similar algometric value for children. The results obtained in our study, together with the aforementioned considerations, suggest that the normal value, below which hyperalgesia should be assessed, is between 1.1 and 2.0 kg/cm^2^.

As in adults, gender is a determining factor for pain. In our study, girls had lower pain threshold values than did boys regardless of the type and weight of the backpack, as previously reported in the literature [30,31,45,46].

Presenting a low pressure pain threshold or a threshold that is lower than the value determined as normal is usually related to the presence of pain in a specific zone [14]; this was shown by the relationships that were found between supraspinatus/cervical pain and shoulder pain, trapezius/cervical pain and middle trapezius/lumbar pain. This could be explained by the pressure exerted by handled backpacks on the anatomical structures studied [16] as well as by the activation of the musculature of these areas in supporting and transporting the weight of the backpack; this activation occurs during the use of both handled and wheeled backpacks [46].

There are some limitations to our study. First, there are different sample sizes in the two groups, and both are small, because the goal of enrolling 30 participants with handled backpacks greater than 10% of their body weight was not achieved. Secondly, the study did not collect any of the additional weight carried by the children, such as musical instruments or sports equipment. In addition, the furniture and the routine of the classes are different in the two schools, a fact that may have affected the results and this study did not consider it. In addition, the participants did not complete the questionnaire by themselves because they did not read or write quickly and perfect, even though they drew the bodymap. Finally, regarding biopsychosocial factors, the authors did not register all of them (expectations, social environment, etc.).

The direction of future studies should include: (1) take shoulder pain into consideration more than lumbar pain; (2) the analysis of biopsychosocial factors; (3) considerate to the new algometry’s values proposed to evaluate child’s hyperalgesia; and (4) a large sample size that takes into account backpack’s discomfort rather than pain, even though no relationship was found between musculoskeletal pain and backpack weight. If we consider, like Dockrell et al. (2015) [11], that carrying a backpack to school can be recommended as a daily exercise, this excessive size makes it difficult for the child to carry the backpack and may even make the child more susceptible to certain types of risks (falls, accidents, etc.). In addition, it is surprising that few children in our study travel to school by walking, despite having less than a 15-min walk. Although our population included few overweight children, this type of exercise would do much to reduce the incidence of childhood obesity [11], especially since the tendency to live a sedentary lifestyle is also increasing [11].

## 5. Conclusions

Most of the children use wheeled backpacks, and only the half of those that use them with handles carry more than 10% of their body weight. The students with handled backpacks weighing more than 10% of their body weight either had no more musculoskeletal pain or had a lower pressure pain threshold than the other participants, although they did report more fatigue. Within this group, no subject suffered back pain; instead, shoulder pain was much more frequent. In addition, all the participants who study in the traditional educational system carry backpacks that exceed 10% of their body weight, and almost all the students in this study carry backpacks that are too large.

This is the first study that analyzes the relationship between the weight of a child’s backpack and the pressure pain threshold.

## Figures and Tables

**Table 1 ijerph-17-02487-t001:** Characteristics of the population and the carrying of backpacks.

Variables	≥10% Group	<10% Group	*p*-Value
Age, years ^1^	8.82 (0.76)	8.78 (0.59)	0.773
Student weight, kg ^2^	30.30 (26.95–34.90)	32.8 (27.20–39.25)	0.189
Student height, cm ^1^	135.83 (7.93)	137.33 (7.86)	0.312
Backpack weight, kg ^2^	5.30 (4.15–7.00)	1.90 (1.48–2.60)	**0.000**
Child-backpack relationship, % ^2^	16.39 (13.15–20.33)	6.29 (4.46–8.15)	**0.000**
Gender ^3^			0.727
Boy	41 (53.2%)	23 (50.0%)	
Girl	36 (46.8%)	23 (50.0%)	
BMI ^3^			0.422
Normal	68 (88.3%)	42 (91.3%)	
Overweight	9 (11.7%)	4 (8.7%)	
Type of backpack ^3^			**0.000**
Handles	20 (26.0%)	44 (95.7%)	
Wheels	57 (74.0%)	2 (4.3%)	
Way of travelling to school ^3^			0.150
Walk	40 (51.9%)	30 (65.2%)	
Car or public transport	37 (48.1%)	16 (34.8%)	
Tiredness in relation to carrying the backpack ^3^			**0.012**
Yes	49 (63.6%)	14 (30.4%)	
No	28 (36.4%)	32 (69.6%)	
Perception that the backpack weighs a lot ^3^			**0.000**
Yes	49 (63.6%)	14 (30.4%)	
No	28 (36.4%)	32 (69.6%)	
Leave the backpack in a locker ^3^			**0.011**
Yes	45 (58.4%)	16 (34.8%)	
No	32 (41.6%)	30 (65.2%)	
Back pain in the family ^3^			0.137
Yes	28 (36.4%)	23 (50.0%)	
No	49 (63.6%)	23 (50.0%)	
Recreational activities			
TV and video games, h ^1^	1.57 (0.82)	1.63 (0.65)	0.680
Physical exercise, h ^1^	4.78 (1.95)	4.24 (2.05)	0.150

^1^ mean (standard deviation); *p*-values were obtained using the t-test. ^2^ median (interquartile interval); *p*-values were obtained using the Mann–Whitney U test. ^3^ percentages; *p*-values were obtained using the chi-square test. *p*-values in bold are statistically significant.

**Table 2 ijerph-17-02487-t002:** Characteristics of pain.

Variables	≥10% Group	<10% Group	*p*-Value
	(*n* = 24)	(*n* = 15)	
Intensity			0.123
Light	5 (20.8%) ^1^	5 (33.3%)	
Moderate	18 (75.0%)	9 (60.0%)	
Severe	1 (4.2%)	1 (6.7%)	
Appearance, months			0.123
<1	2 (8.3%)	2 (13.3%)	
1–3	10 (41.7%)	1 (6.7%)	
3–6	8 (33.3%)	7 (46.7%)	
>6	4 (16.7%)	5 (33.3%)	
Frequency, days/week			**0.008**
Frequently (4–5)	0 (0.0%)	0 (0.0%)	
Sometimes (2–3)	1 (4.2%)	3 (20.0%)	
Occasionally (1)	2 (8.3%)	6 (40.0%)	
Rarely (<1)	21 (87.5%)	6 (40.0%)	
Duration, hours			0.381
<1	19 (79.1%)	9 (60.0%)	
1–2	4 (16.7%)	4 (26.7%)	
2–6	1 (4.2%)	2 (13.3%)	
6–12 or more	0 (0.0%)	0 (0.0%)	
Missed school			0.385
Yes	0 (0.0%)	1 (6.7%)	
No	24 (100.0%)	14 (93.3%)	
Visited doctor			0.908
Yes	6 (25.0%)	4 (26.7%)	
No	18 (75.0%)	11 (73.3%)	

^1^ percentages; *p*-values were obtained using the chi-square test. ***p*-values** in bold are statistically significant.

**Table 3 ijerph-17-02487-t003:** Areas of pain.

Variables	≥10% Group	<10% Group	*p*-Value
Cervical	7 (87.5%)^1^	1 (12.5%)	0.121
Interscapular	9 (64.3%)	5 (35.7%)	0.792
Lumbar	2 (28.6%)	5 (71.4%)	0.085
Shoulders	27 (69.2%)	12 (30.8%)	0.477

^1^ percentages; *p*-values were obtained using the chi-square test.

**Table 4 ijerph-17-02487-t004:** Pressure pain threshold in kg/cm^2^.

Variables	≥10% Group (*n* = 77)	<10% Group (*n* = 46)	*p*-Value
Upper trapezius right	1.20 ^1^ (1.00–1.50) ^2^	1.10 (1.00–1.40)	0.771
Upper trapezius left	1.10 (0.00–1.40)	1.05 (0.00–1.40)	0.816
Middle trapezius right	1.40 (1.10–1.80)	1.45 (1.20–1.90)	0.619
Middle trapezius left	1.30 (1.00–1.80)	1.50 (1.10–1.90)	0.672
Levator scapulae right	0.00 (0.00–1.00)	0.00 (0.00–1.10)	0.583
Levator scapulae left	0.00 (0.00–1.00)	0.00 (0.00–1.00)	0.482
Supraspinatus right	1.40 (1.00–1.50)	1.20 (1.10–1.50)	0.623
Supraspinatus left	1.20 (1.00–1.50)	1.10 (1.00–1.50)	0.800

^1^ median and ^2^ interquartile interval; *p*-values were obtained using the Mann–Whitney U test.

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
