# Peer review of "Relationship between School Backpacks and Musculoskeletal Pain in Children 8 to 10 Years of Age: An Observational, Cross-Sectional and Analytical Study"

_ijerph, 2020, doi:10.3390/ijerph17072487_

Round 1

Reviewer 1 Report

The study is well designed and presents important information related to school backpacks in 8-10-year group children.

I have some comments.

Materials and methods. The prelavence of pain is missing a reference (line 84).

Results. Include TV and video games (line 198). How was the cardiovascular demand assess?

Discussion. The limitations should be more accurately presented. The indications for the future research may be described as recommendations.

Conclusions. The conclusions must be rephrased pointing out the main findings of the study.

Reviewer 2 Report

Thank you for submitting this interesting and very relevant article.  The abstract would benefit from a summary and/or  recommendations statement at the end and  also reference to the recommended maximum in the body of the text of the abstract.  

Line 27  'and it seems' would be better 

Line 43 with cervical and shoulder pain  

Avoid excessive use of ; as this makes the text disjointed.

Line 45 clarify it is the weight of the back pack 

Line 61 clarify what the other group did i.e. was it carried on their back ?

Line 72 method is the correct work not methodologies  

Line 80  you use the word collaborate  do you mean co-operate?

Line 82 this is the first mention or pressure pain threshold I suggest you refer to it on the background

Line 88/89 you refer to activity level in the abstract but not here. 

You need to refer to reliability and validity  of all the measures you are taking 

Line 96 Re word the information was collected by Researcher (supply initials)

Provide justification of choice  or areas for measuring the pressure pain threshold 

Line 120-122 you need to name the questionnaires 

Line 124-26  I would expect reference to to an information sheet 

Line 135 Detail what anthropometric  data was collected 

Line 138 is this correct that the student did not complete the questionnaires -that the student was interviewed and  then the researcher completed the questionnaires  ? This poses issue regrading bias ensure this is in the limitations if my interpretation is correct 

Where there any attempts at standardisation of the placement of the algometer ?

Line 188 justify why the majority of back packs were too large --does this mean too heavy  ? needs clarification 

The Discussion covers all the  main areas 

Round 2

Reviewer 1 Report

All my comments were answered.

Author Response

Thank you for the corrections. 

Reviewer 2 Report

Abstract :

Line 10 Should be in present not past tense.   Back pain in children is a reality  and various factors are involved  

Line  14 should say and their physical activity 

Line 15 would be better saying   The results indicated that all backpacks .....

Line 18 19 you need to indicate where the Pressure pain thresholds were collected from -just give general details 

Line 22 It's not clear what the term biopsychsocial refers to in this context as the study does not really gather data on this in totality.  

Main paper

Line 48/49 this is an odd place to reference to Pressure pain thresholds. I suggest you need to find a paper that does use Algometry to measure PPT and refer to it. It may not be in the same context i.e. backpacks  and children  but that is not a problem  it just needs to be in a study related back pain.

Line 51 Starts with For this reason --however the text following does't necessarily link to the last sentence of previous paragraph.  If you place the section on algometry somewhere else it may link better.

Line 54 55 say muscles chosen for examination but doesn't say in what way-

I suggest delete these 4 words and add tot eh section on algometry

Line 68-71 This is a repetitive and longwinded  collection of phrases- re write into a more succinct clear hypotheses

Line 111 Sentence start with Has. please clarify what has and reword.

Line 130 suggest remove brackets and say lifestyle habits including mode of travel to School....

Line 162 Reference needed for the use of the muscle belly as the point to measure PPT.

Line 206 Reference needed to support statement about the size of back pack

Line 398 399 Point 3 referring algometery doesn't make sense please rephrase 

Lin3 399 I suggest deleting point 4 ore rephrase to say  if the study/topic  needs doing again it needs to be conducted with larger sample size 

Line 409. You say very few children carry their back packs with handles but you need to clarify how they do carry them 
